# Whole-Exome Sequencing Reveals High Mutational Concordance between Primary and Matched Recurrent Triple-Negative Breast Cancers

**DOI:** 10.3390/genes14091690

**Published:** 2023-08-25

**Authors:** Jaspreet Kaur, Darshan S. Chandrashekar, Zsuzsanna Varga, Bettina Sobottka, Emiel Janssen, Khanjan Gandhi, Jeanne Kowalski, Umay Kiraz, Sooryanarayana Varambally, Ritu Aneja

**Affiliations:** 1Department of Biology, Georgia State University, Atlanta, GA 30303, USA; jaspreet070781@gmail.com; 2Department of Pathology—Molecular and Cellular, University of Alabama at Birmingham, Birmingham, AL 35233, USA; dshimogachandrasheka@uabmc.edu (D.S.C.); svarambally@uabmc.edu (S.V.); 3Department of Pathology and Molecular Pathology, University Hospital Zurich, 8091 Zurich, Switzerland; zsuzsanna.varga@usz.ch (Z.V.); annabettina.sobottka-brillout@usz.ch (B.S.); 4Department of Pathology, Stavanger University Hospital, Health Stavanger HF, 4068 Stavanger, Norway; emiel.janssen@uis.no (E.J.); umay.kiraz@sus.no (U.K.); 5Winship Cancer Institute, Emory University, Atlanta, GA 30322, USA; gandhikhanjan@gmail.com; 6Livestrong Cancer Institutes, Dell Medical School, The University of Texas at Austin, Austin, TX 78712, USA; jeanne.kowalski@austin.utexas.edu; 7Department of Clinical and Diagnostic Sciences, School of Health Professions, University of Alabama at Birmingham, Birmingham, AL 35294, USA

**Keywords:** triple negative breast cancer, tumor mutational burden, whole exome sequencing

## Abstract

Purpose: Triple-negative breast cancer (TNBC) is a molecularly complex and heterogeneous breast cancer subtype with distinct biological features and clinical behavior. Although TNBC is associated with an increased risk of metastasis and recurrence, the molecular mechanisms underlying TNBC metastasis remain unclear. We performed whole-exome sequencing (WES) analysis of primary TNBC and paired recurrent tumors to investigate the genetic profile of TNBC. Methods: Genomic DNA extracted from 35 formalin-fixed paraffin-embedded tissue samples from 26 TNBC patients was subjected to WES. Of these, 15 were primary tumors that did not have recurrence, and 11 were primary tumors that had recurrence (nine paired primary and recurrent tumors). Tumors were analyzed for single-nucleotide variants and insertions/deletions. Results: The tumor mutational burden (TMB) was 7.6 variants/megabase in primary tumors that recurred (*n* = 9); 8.2 variants/megabase in corresponding recurrent tumors (*n* = 9); and 7.3 variants/megabase in primary tumors that did not recur (*n* = 15). *MUC3A* was the most frequently mutated gene in all groups. Mutations in *MAP3K1* and *MUC16* were more common in our dataset. No alterations in *PI3KCA* were detected in our dataset. Conclusions: We found similar mutational profiles between primary and paired recurrent tumors, suggesting that genomic features may be retained during local recurrence.

## 1. Introduction

Triple-negative breast cancer (TNBC), a highly aggressive breast cancer subtype that lacks the expression of estrogen receptor (ER), progesterone receptor (PR), and human epidermal growth factor receptor 2 (HER2), accounts for 10–20% of all breast cancers [1,2]. Most TNBCs are high-grade, poorly differentiated carcinomas with high proliferation rates. TNBCs are challenging to treat due to their highly heterogeneous nature, rapid proliferative capabilities, chemoresistance, and high rates of metastasis to distant organs and tumor recurrence [3,4]. Although only 5% of TNBC patients are diagnosed with de novo metastatic disease, most of the patients typically experience relapse following treatment with curative intent [5,6]. TNBCs typically metastasize to the brain, liver, and lungs [7]. The prognosis of patients with metastatic TNBC is poor, and the majority of deaths among TNBC patients are due to progressive metastatic disease. Unlike other breast cancer subtypes, TNBC lacks clinically available tests to predict the risk of distant metastasis or disease relapse. Additionally, there are no metastasis-specific biomarkers to identify and treat patients at high risk of metastasis. The identification of molecules and pathways that drive TNBC metastasis may uncover distant metastasis-specific biomarkers. Hence, an in-depth understanding of the biology of metastatic dissemination may improve survival outcomes in patients with TNBC.

High-throughput next-generation sequencing techniques such as whole-exome sequencing (WES) have provided enormous insights into the genomic landscapes of several tumor types, leading to the identification of new druggable targets and the definition of new tumor subtypes and shedding light on the heterogeneity of many tumors [8,9]. In particular, through target enrichment, WES represents a cost-effective strategy for the identification of mutations in the protein-coding exons of the human genome. Essentially, knowledge of alterations in the coding regions of all genes in the genome may guide immediate treatment choices and strengthen therapeutic discovery efforts. Furthermore, the reduced costs and increased practical availability of tumor genomic profiling has generated ample opportunities to test “the precision medicine” hypothesis in clinical oncology. However, the approach suffers from a number of challenges that limit its application for widespread clinical WES implementation. The foremost is the rapid generation of high-quality WES data from archival FFPE tissue. Next is the ability to clinically interpret WES data for prospective use, which could maximize clinical and biological explorations. Overcoming these limitations would allow the rigorous assessment of the value of WES to guide clinical decision making [10].

The clinical management of metastatic cancer often relies on actionable molecular targets derived from primary tumors [11]. Potential genomic discordances in the molecular profiles of primary tumors and metastatic lesions are therapeutically relevant. Discordance of actionable molecular targets between primary tumors and metastatic recurrence can result in the non-optimal treatment of metastatic disease or cause unnecessary side effects [12,13,14]. Although the mutational landscape of primary breast tumors has been extensively analyzed, the mutational profiles of metastatic or recurrent breast tumors remain elusive [15,16]. Paired analyses of primary and metastatic tumors are pivotal for the optimal management of metastatic disease because (i) spatial and temporal differences might exist between primary tumors and matched metastatic lesions [17]; (ii) disseminating metastatic cells from primary tumors can activate specific transcriptomic programs to colonize and adapt to new tissue microenvironments; and (iii) additional molecular changes may be acquired by metastatic tumors due to therapeutic interventions, such as adjuvant chemotherapy [18]. Hence, the molecular profiling of primary tumors and matched metastatic lesions could facilitate the identification of actionable metastasis-specific targets. In this study, we performed whole-exome sequencing (WES) of nine matched primary and recurrent TNBC tumors to compare their mutational profiles and identify molecular alterations associated with metastatic progression in TNBC. We also performed WES of 15 primary tumors that remained recurrence-free to compare the mutational landscapes of primary TNBC tumors that recurred to those that did not recur.

## 2. Materials and Methods

### 2.1. Patients and Patient Samples

Formalin-fixed paraffin-embedded (FFPE) primary and matched recurrent tumor tissue (*n* = 35 samples) from 26 patients with TNBC was obtained from the University Hospital Zurich, Switzerland (Table 1). Fifteen of the 35 samples were tissue from primary tumors that did not recur, and 11 samples were primary tumors that recurred (9 paired primary and recurrent tumors). Of the 9 recurrent tumors, 7 (78%) were lymph node recurrences, and 2 (22%) were soft tissue and intramammary recurrences.

All 9 patients with paired primary and recurrent samples were diagnosed with recurrences (lymph node or other sites) after treatment for early or advanced breast cancer. The median age at diagnosis was 55 years. The median follow-up time was 3 years for patients with recurrence and 2.4 years for patients without recurrence. The clinicopathological characteristics of the patients are summarized in Table 1. The status of ER, PR, and HER2 was evaluated using immunohistochemistry (IHC). Because of the retrospective study design, we did not have access to blood samples or matched non-tumor tissue for these patients. Therefore, we performed variant filtering by frequency in a healthy population to exclude potential germline mutations. Previous comparisons of germline variants between unrelated individuals have shown that germline variants can be used as an effective filter, obviating the need for sequence-matched tumors and normal tissue [19,20]. The study protocol was approved by the Institutional Review Board and was in compliance with material transfer guidelines and data use agreements between Georgia State University and the University Hospital Zurich, Switzerland. The study was conducted in accordance with International Ethical Guidelines for Biomedical Research involving human subjects. Written informed consent was obtained from all the participants.

### 2.2. WES and Variant Calling

Hematoxylin and eosin (H&E) slides were prepared for all samples, and the tumor content was assessed by a pathologist. Genomic DNA (gDNA) was extracted using the NucleoSpin DNA FFPE Kit (Macherey-Nagel, Düren, Germany), and its concentration and purity were measured using NanoDrop and Qubit (ThermoScientific, Waltham, MA, USA). DNA electrophoresis using 2200 Tapestation (Agilent Technologies, Inc., Santa Clara, CA, USA) was conducted to confirm gDNA purity and concentration. One of the challenges of working with FFPE samples is that DNA extracted from this tissue is often of limited quantity. DNA yields from FFPE tissue samples might be insufficient for standard next-generation sequencing protocols. Despite this limitation, several studies have successfully sequenced samples starting with inputs as low as 10 ng [21,22]. The DNA available as input in our study ranged from 0.06 µg to 5.6 µg. Additionally, the majority of the samples sequenced (34/35) had a DNA integrity number ≥ 3 (Appendix A). The average coverage depth was 100×.

WES libraries were prepared using the SureSelect V6 exome kit from Agilent (Santa Clara, CA, USA), following the manufacturer’s instructions. The resulting libraries were sequenced on a NextSq 500 system (Illumina, San Diego, CA, USA) according to the standard operation protocol. The sequence quality of the resulting paired-end 150-nucleotide reads was assessed using FastQC [23].

Reads were trimmed using Trim-Galore to remove adapter sequences and low-quality sequences. Trimmed reads were mapped to the human reference genome GRCh38 using the BWA software [24]. After alignment, all samples were preprocessed according to the Genomics Analysis Toolkit (GATK) germline variant calling best practice workflow [25]. Germline variant calling was performed on all the samples using GATK HaplotypeCaller followed by VariantRecalibrator and ApplyVQSR.

GATK output files (VCF files) were screened for high-quality variants using SnpEff [26]. Mutants with a ‘PASS’ filter or no filter and a quality score of 30 or more were considered for downstream analysis. VCF files related to the same tumor type were merged using BCFtools, and merged VCF files were annotated using ANNOVAR [27]. ANNOVAR was used to filter out mutations with a minor allele frequency (MAF) of 0.01 in the “exac03”, “esp6500siv2”, and “gnomad_exom” germline databases. Thereafter, ANNOVAR was used to annotate variants and screen each variant using the LJB* databases. All single-nucleotide variants (SNVs) were scored using the SIFT, PolyPhen2 HDIC, LRT, MutationTaster, MutationAssesor, and FATHMM tools to predict the effect of mutation on each corresponding protein. SNVs that were found to have deleterious effects by at least 3 tools were selected for further analysis. This process ensured that common germline mutations and mutations with no adverse effect on the protein were excluded. Using the maftools R package, we analyzed the mutational landscapes of primary tumors and metastatic tumors separately and performed a comparative mutational analysis between tumor groups [28].

## 3. Results

### 3.1. Mutational Landscape of TNBC

The WES of 35 samples from 26 patients with TNBC revealed a total of 33,853 variants. Among the nine paired primary and recurrent tumor samples, the median number of variants per sample was 732 for primary tumors and 781 for the corresponding recurrent tumors. The median number of SNVs was 431 in primary tumors and 456 in matched recurrent tumors. We detected a total of 219–445 indels in primary tumors and 228–2089 indels in the corresponding recurrent tumors (Appendix A). We used variant data to calculate the tumor mutational burden (TMB) for each sample, which was defined as the number of mutations per megabase of the human genome sequenced. The median TMB in the nine primary tumors was 7.6 variants per megabase, and the median TMB in matched recurrent tumors was 8.2 variants per megabase. The TMB observed in our dataset is similar to the previously reported average TMB of 7.3 variants per megabase in the Thai TNBC dataset [29].

We found a significant overlap in the genes mutated in the nine primary and recurrent tumors; however, a few genes were mutated only in primary or recurrent tumors (Figure 1). Mutations in *ATXN3*, *CAMKK2*, *L00134391*, *MAML2*, *NCOR2*, *RPL14*, *TRAK1*, and *ZAN* were enriched only in primary tumors. On the other hand, mutations in *COL17A1*, *LAMC3*, and *MMP27* were restricted to recurrent tumors (Figure 1). Among the genes mutated only in recurrent tumors, *COL17A1* has been shown to prevent breast cancer cell invasion and migration [30]. Based on the number of mutations found in each gene, we identified the top 10 most mutated genes in both primary and matched recurrent tumors. *MUC3A* was the most frequently mutated gene in both groups (Appendix A).

We also compared the mutational profiles of primary tumors with recurrence (*n* = 11) to those of primary tumors without recurrence (*n* = 15). The median variant number per sample was 696 for primary tumors without recurrence and 764 for primary tumors that recurred. In total, 219–1171 indels were detected in primary tumors that recurred, and 208–389 indels were detected in primary tumors that did not recur. The median number of SNVs was 431 in primary tumors that recurred and 398 in primary tumors that remained recurrence-free (Appendix A). We found that the TMB in recurrent-free primary tumors was 7.3 variants per megabase.

Next, we compared the genes that were altered in all the samples in both groups. We found that the genes *C9orf147*, *FMO2, FOLR3, GPR33, L00134391, MUC3A, MUC12, POLR2A, RPL14,* and *USF3* were mutated in both the primary tumors that recurred and in those that did not recur. *MUC3A* remained the most mutated gene in primary tumors without recurrence (Appendix A).

### 3.2. Frequency of Recurrent Gene Mutations in TNBC

cBioportal was used to identify breast-cancer-specific genes; *PI3KCA*, *TP53*, *PTEN*, *GATA3*, *SYNE1*, *MAP3K1*, *MUC16*, and *CDH1* were the genes with the highest mutational frequencies in the TCGA and METABRIC cohorts. Additionally, we analyzed mutational alterations in TNBC-associated risk genes, including *BRCA1* and *BRCA2*. We compared the frequency of recurrent gene mutations in matched primary and recurrent tumor pairs (Figure 2). *PTEN* is a negative regulator of the PI3K pathway, and mutations in *PI3KCA* and *PTEN* are often mutually exclusive [31,32,33]. Although *PI3KCA* was not mutated in any of the primary or recurrent tumors, three of nine primary and recurrent tumors and one primary–recurrent tumor pair (P4-R4) harbored a mutation in *PTEN*.

*MAP3K1* mutations are more frequent in hormone-receptor-positive (HR+) breast cancer than in TNBC [34]. However, in our dataset, *MAP3K1* was mutated in all primary TNBC tumors and their matched recurrent tumors. Although primary tumors had multiple mutations in *MAP3K1*, all recurrent tumors had in-frame *MAP3K1* deletions. Interestingly, mutations in *CDH1* and *GATA3* were observed in only two primary TNBC tumors (P6 and P9); similar alterations in these genes were missing in their paired recurrent samples. In addition, no mutations in *CDH1* and *GATA3* were observed in any of the other recurrent tumors. Although 8/9 primary tumors and 6/9 recurrent tumors harbored an alteration in *MUC16*, five out of nine matched primary–recurrent pairs (P1-R1, P3-R3, P5-R5, P7-R7, and P8-R8) had *MUC16* alterations. Similarly, *SYNE1* was mutated in around 6/9 recurrent tumors, 4/9 primary tumors, and only two matched primary–recurrent pairs (P6-R6 and P7-R7). *TP53* was mutated in three primary and three recurrent tumors; however, only two matched primary–recurrent pairs (P1-R1 and P6-R6) shared a *TP53* mutation. Although *BRCA2* was altered in the primary tumor P3, this mutation was not observed in the corresponding recurrent tumor. Only one primary–recurrent tumor pair (P1-R1) exhibited *BRCA2* mutations. No pair-wise alterations were observed in *BRCA1*; one primary tumor (P2) and one recurrent tumor (R7) harbored a frameshift deletion and multiple *BRCA1* mutations, respectively.

Next, we assessed the frequency of recurrent gene mutations in primary tumors with and without recurrence (Figure 3). We found that mutations in *PI3KCA* and *PTEN* were mutually exclusive in the four primary tumors that recurred. Primary tumors without a recurrence (except for P10′) had no mutations in *PTEN* or *PI3KCA*. Intriguingly, *MAP3K1* was mutated in all primary tumors except for one (P15′). No alterations in *CDH1* and *GATA3* were observed in the primary tumors that remained recurrence-free, although both of these genes were mutated in at least one of the primary tumors that later had a recurrence. *TP53* was altered in 9/15 primary tumors that did not recur and only 4/11 primary tumors that recurred. *MUC16* mutations were detected in most primary tumors; 10/11 primary tumors that recurred and 9/15 primary tumors in those that remained recurrence-free harbored *MUC16* mutations. Only 2 and 3 out of 11 primary tumors that recurred exhibited *BRCA1* and *BRCA2* mutations, respectively; *BRCA1* and *BRCA2* were mutated in 2/15 primary tumors that did not recur. 

## 4. Discussion

In this study, we investigated the mutational landscapes of primary TNBC tumors and their matched recurrent tumors and compared the mutational profiles of primary tumors that had recurrence to those that remained recurrence-free. Mutational analysis in TNBC patients remains fertile ground for the search for actionable targets owing to the inherent heterogeneous nature of the disease and the varied outcomes seen in the patients. Notably, genomic alterations were observed in all the samples investigated. Primary TNBC tumors and matched recurrent tumors showed similar mutational profiles, evidenced by the similar TMB and number of variants per sample. There was also an overlap in gene mutations between primary and matched recurrent tumors. Discordance in gene alterations between primary and matched recurrent TNBC tumors can be attributed to temporal and spatial differences between primary and recurrent lesions and neoadjuvant chemotherapy received by four of the nine (~44%) patients with matched primary and recurrent tumors. The neoadjuvant chemotherapy may have led to the acquisition of mutations in recurrent tumors that were initially absent in the corresponding primary tumors. 

The high concordance in the mutational landscapes observed in our dataset is in line with previous studies showing similar genomic alterations in primary tumors and metastatic or recurrent tumor samples [15,35]. For instance, a study by Moreno et al. found that 85.5% of variants in primary tumors were also present in metastatic tissue [36]. Similarly, a study by Roy-Chowdhuri et al. also found 77% concordance between matched primary and metastatic breast tumors [37]. Varying degrees of genomic concordance between primary tumors and their corresponding metastatic tumors have been previously reported, suggesting that primary tumors harbor genetic alterations essential for successful metastatic dissemination [15,35]. These findings also suggest that the mutational signatures of primary tumors can serve as a proxy for cells that ultimately participate in metastatic dissemination and are responsible for tumor recurrence and disease relapse.

The identification of consistent mutational profiles between primary and corresponding recurrent tumors can aid in the identification and development of novel diagnostic and therapeutic targets in TNBC. The largely similar profiles of primary and recurrent tumors may prove useful in clinical research investigating global mutational changes during tumor progression or treatment responses, and for the clinical management of TNBC patients. For instance, the detection of mutations in immune response genes in TNBC allows for the prediction of the efficiency of response to immune checkpoint inhibitors or anti-PDL1 drugs [38]. Similarly, mutations in the genes that were altered in both primary and recurrent tumors (Figure 1) may allow for the tailoring of the targeted therapy and prediction of the response to treatment. The high recurrence risk is one of the main problems in the clinical management of TNBC. A comparison of the mutational profiles between the primary tumors that recurred compared to the ones that did not (Figure 3) could lead to the development of mutational signatures that can help to predict recurrence. From this point of view, the unique mutational signatures that can prognosticate patients into high- and low-recurrence groups could be valuable. 

Non-recurrent primary tumors also harbored a similar number of variants as the primary tumors that metastasized. *MUC3A* was the most frequently mutated gene both in primary tumors with recurrence and in those that did not have recurrence. The differences in the frequency of gene mutations between the primary tumors that recurred and those that did not have recurrence could be attributed to differences in follow-up times between the comparison groups. *MUC3A* was also the top mutated gene both in primary TNBC tumors and in recurrent tumors. *MUC3A* encodes mucin 3A, a protein that belongs to the family of mucins, which are large glycoproteins expressed in various epithelial and malignant cells. The abnormal expression or glycosylation of mucins results in alterations in cell growth, differentiation, adhesion, and invasion and has been implicated in the development of neoplasms, including breast cancer [39,40]. Rakha et al. found that *MUC3* was expressed in 91% of invasive breast cancer samples and that its expression was significantly associated with the lymph node stage, a poor Nottingham prognostic index (NPI), a high grade, and an increased risk of local recurrence [41]. *MUC16*, a gene encoding another mucin family member, was also mutated in most tumors in our dataset. Interestingly, previous findings suggest a role for *MUC16* in promoting metastasis, therapy resistance, and disease progression in multiple malignancies, including breast cancer [42,43,44,45,46].

Previous next-generation sequencing studies have revealed the distinct mutational spectrum of TNBC. *TP53* is the most commonly mutated gene (up to 80%) [47], whereas *PI3KCA* has the lowest mutational frequency in TNBC (approximately 9%) [48,49]. Consistently, we found no *PI3KCA* mutations in primary TNBC tumors (irrespective of their recurrent status) and matched recurrent tumors. The frequency of *PI3KCA* mutations is substantially higher in hormone-receptor-positive breast cancer compared with the TNBCs as a whole. Reports have suggested an association between TNBC molecular subtypes and alterations in the PI3K pathway. Specifically, it has been reported that *PI3KCA* mutations are more common in luminal TNBC and found in up to 40% of androgen-receptor-positive TNBCs [50,51,52]. Since we did not have information about the molecular subtypes of the TNBCs used for the study, it is possible that the luminal subtype was under-represented or absent in this cohort owing to the small sample size. Nonetheless, the dominance of *PI3KCA* mutations in specific TNBC subsets suggest the potential for targeted therapy for *PI3KCA*-mutant TNBCs. The frequency of *TP53* mutations varied between primary and recurrent tumors in our dataset. A recent study showed that patients with TNBC harboring mutant *TP53* and wild-type *PI3KCA* could benefit from immune checkpoint inhibitors (ICIs) [53].

*MAP3K1* (or *MEKK1*) encodes a serine/threonine kinase that regulates the activity of various kinases regulating cell proliferation, migration, survival, and death [34,54]. Comprehensive genomic analyses revealed multiple alterations in *MAP3K1* in different cancer types, including ER+ breast cancer. Except for *TP53* and *PI3KCA*, most of the significantly mutated genes in non-TNBCs are rarely mutated in TNBC [34]. However, in our dataset, primarily composed of TNBC samples, we found a striking mutation pattern for *MAP3K1*, which was altered in all the samples except for one primary tumor (P15′). In addition to commonly mutated genes, infrequently mutated genes contribute to the mutational landscape of TNBC and may serve as actionable targets.

*BRCA1* and *BRCA2* are tumor suppressor genes involved in DNA repair and genome integrity. Somatic or germline *BRCA1* or *BRCA2* mutations are found in 10% to 40% of patients with TNBC. Often, high-grade breast cancer and TNBCs show somatic mutations or abnormal *BRCA1*/*BRCA2* expression [55,56,57,58]. The varied prevalence of *BRCA1*/*BRCA2* mutations in this study could be attributed to a number of factors, including the age at diagnosis, menopausal status, ethnicity, and therapy. Patients harboring mutations in genes involved in DNA damage repair, including *BRCA1* and *BRCA2*, may benefit from platinum-based chemotherapy or poly(ADP-ribose) polymerase (PARP) inhibitors [59,60].

Our study had several limitations. First, the cohort size was small; however, this dataset represented a unique resource, as it included matched primary and recurrent tumor samples from the same patient. It is challenging to obtain matched sets of primary and recurrent tumors because biopsy samples of recurrent tumors are rare. Second, cases were chosen based on sample availability, inadvertently introducing selection bias. Third, although the incidence of TNBC is higher in young patients than in older individuals, more than half of our samples were obtained from older (≥50 years) patients. Nevertheless, the comparison of TMB data from our dataset with previously published literature could have been limited by differences in cohort, study design, and data analysis methods. Moreover, the median follow-up time of 3 years was not long enough to identify clinically relevant relationships between genomic alterations and clinical outcomes. Additionally, we used whole tumor samples instead of laser-capture microdissection samples [61]. Although great care was taken to reduce contamination from stromal and immune components, tumor microenvironment components that remained may have influenced our findings [62].

Next-generation sequencing technologies enable the characterization of the mutational and transcriptomic profiles of primary and metastatic tumors. Genome-wide comparisons of gene expression profiles in paired samples can aid in the detection of mutations that drive malignancy and perturbed genes and pathways that promote metastasis. Massively parallel sequencing techniques enable the comparison of the global gene expression profiles of matched primary and metastatic/recurrent TNBC tumors. Additionally, quantitative and spatial proteomics approaches can aid in strengthening the next-generation sequencing study findings. Ongoing studies in our laboratory aim to recruit large cohorts of patients with metastatic disease and integrate various omics platforms to map the complex genomic and proteomic profiles of primary and recurrent and/or metastatic tumors from the same patient. We envision that an integrative approach will provide a holistic view of the complexity associated with the metastatic dissemination of cancer cells. This in turn will ultimately lead to the refinement of our understanding of metastatic TNBC and the identification of metastasis-specific biomarkers and improve outcomes in TNBC patients by elucidating the multi-level alterations during metastatic disease progression.

## 5. Conclusions

Metastatic TNBCs are particularly aggressive, and the lack of actionable molecular alterations makes metastatic TNBC a challenging disease. Although the mutational landscapes of primary TNBC have been extensively characterized, analyses of paired metastatic lesions are scarce. In this study, we found that the primary tumors and metastatic lesions had similar mutational landscapes, suggesting that the primary tumor could serve as a surrogate for the detection of disseminated cancer cells and locoregional recurrences. However, we also found that mutations in *COL17A1*, *LAMC3*, and *MMP27* were enriched only in recurrent tumors. This finding indicates that the TNBCs in our cohort followed a mixed model of tumor evolution and that recurrent tumors consisted of both clones that disseminated early and late from primary tumors during tumor evolution. *MUC3A* was the most frequently mutated gene in the dataset. This study led to the identification of previously unexplored, metastasis-specific, actionable targets (i.e., *MUC3A*, *COL17A1*, *LAMC3*, and *MMP27*) that can be further validated and developed for the management of metastatic TNBC.

## Figures and Tables

**Figure 1 genes-14-01690-f001:**
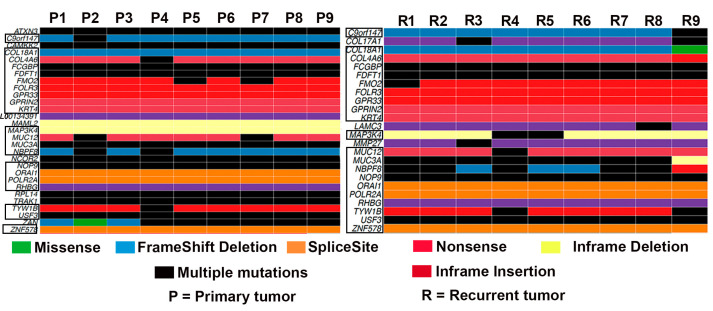
Oncoplot showing the frequency of gene mutation in all the primary tumors and matched recurrent tumors (*n* = 9). Genes that are mutated in both groups are shown in the boxes.

**Figure 2 genes-14-01690-f002:**
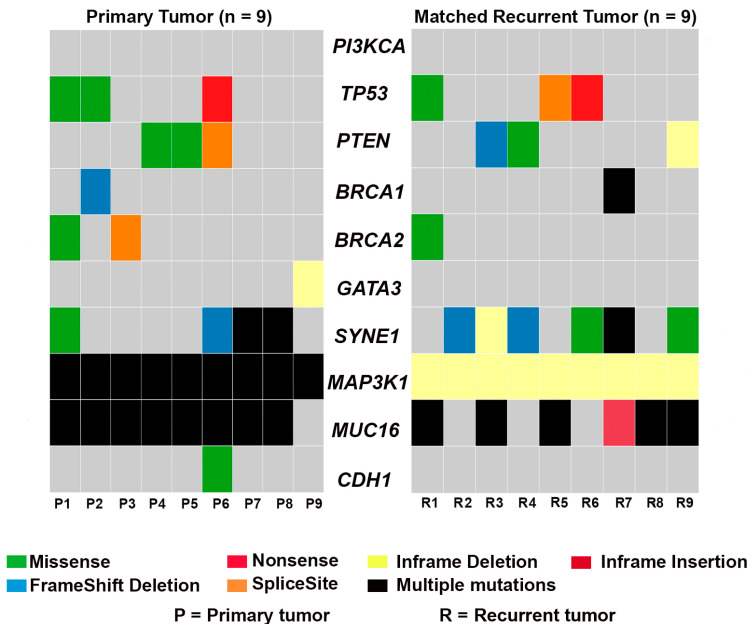
Oncoplot showing the frequency of recurrent gene mutations in paired primary and recurrent tumors.

**Figure 3 genes-14-01690-f003:**
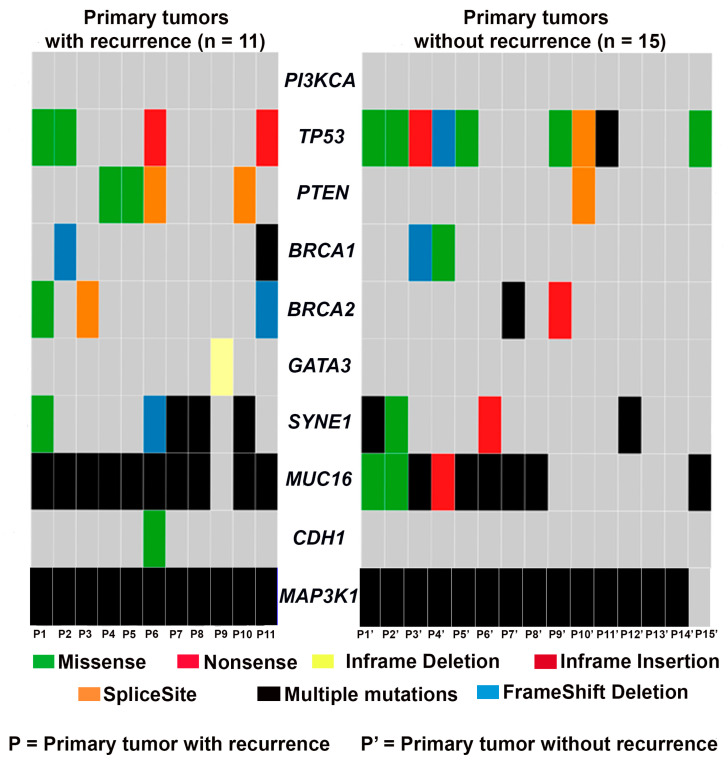
Oncoplot showing the frequency of recurrent gene mutations in primary tumors that recurred (*n* = 11) and those that remained recurrence-free (*n* = 15).

**Table 1 genes-14-01690-t001:** Descriptive statistics for the clinicopathological characteristics of patients with TNBC.

Baseline Characteristics	Recurrence (Lymph Node or Other)	Recurrence-Free	*p*-Value
Patient Age, n (%)			
20–29	0 (0.00)	1 (6.70)	0.877
30–39	2 (18.18)	1 (6.70)	
40–49	2 (18.18)	3 (20.00)	
50–59	4 (36.36)	6 (40.00)	
60–69	2 (18.18)	1 (6.70)	
70+	1 (9.09)	3 (20.00)	
Tumor Grade, n (%)			
I	0 (0.00)	0 (0.00)	0.492
II	0 (0.00)	2 (13.33)	
III	11 (100.00)	13 (86.67)	
Histological Type, n (%)			
NST (Ductal)	8 (72.72)	13 (86.67)	0.521
(With Secretory Differentiation)	1 (9.09)	0 (0.00)	
Apocrine	1 (9.09)	2 (13.33)	
Metaplastic	1 (9.09)	0 (0.00)	
Survival Status, n(%)			
Alive	6 (54.54)	11 (73.33)	0.418
Dead	5 (45.45)	4 (26.67)	

## Data Availability

The data presented in this study are available in this article (and Supplementary Material).

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
