# Peer review of "Whole-Exome Sequencing Reveals High Mutational Concordance between Primary and Matched Recurrent Triple-Negative Breast Cancers"

_genes, 2023, doi:10.3390/genes14091690_

Round 1
Reviewer 1 Report
The whole-exome sequencing method was used to study the mutational landscape of primary TNBC tumors and their matched recurrent tumors and compared the mutational profiles of primary tumors that had a recurrence to those that remained recurrence-free to achieve the purpose of studying the genetic characteristics of TNBC. In this study, it was demonstrated that the mutation spectrum between primary tumors and paired recurrent tumors is similar, and the most common mutation genes in primary and paired recurrent TNBC tumors were revealed. But before considering publication, there are some issues that must be addressed here. If these problems can be solved, this paper will be even better.
1.The significance of this paper is not expounded sufficiently. The author needs to highlight this paper's innovative contributions.
2.The background section is not comprehensive enough. It is recommended to cite more previous research to introduce the characteristics of primary TNBC and matched recurrent TNBC, as well as the advantages of WES.
3.On page 2, lines 67-68“The status of ER, PR, and HER2 was evaluated using immunohistochemistry (IHC).”Can the authors provide the results of IHC here?
4.The research purpose of this paper needs to be further described.
5.If the 'Conclusion' section can be added, it will be more beneficial for readers to understand the article.
Reviewer 2 Report
Suggestions and Comments for Major Revision:
Introduction:
a. Provide a brief background on triple-negative breast cancer (TNBC), emphasizing its clinical significance, high risk of metastasis, and poor prognosis. This will contextualize the importance of studying the molecular mechanisms underlying TNBC metastasis.
Methods:
a. Clarify the criteria used for selecting the 26 TNBC patients included in the study. Explain the rationale behind choosing these particular patients and provide information on any relevant clinical characteristics or treatment history that might influence the interpretation of the results.
b. Elaborate on the specific techniques and protocols employed for whole-exome sequencing (WES) analysis. Include information regarding the coverage depth, quality control measures, and any bioinformatics pipelines utilized for variant calling and annotation. Additionally, mention any specific databases or references used to interpret the functional impact of the identified variants.
Results:
a. Provide a comprehensive description of the mutational landscape observed in the primary TNBC tumors and paired recurrent tumors. Include details on the total number of somatic mutations, their distribution across genomic regions, and the specific functional consequences of the identified variants. Consider presenting this information in tables or figures to enhance clarity.
b. Discuss the observed differences in the mutation frequencies of MUC3A, MAP3K1, and MUC16 between the primary and recurrent tumors. Are these differences statistically significant? If so, provide relevant statistical analyses to support the findings.
c. While it is noteworthy that no alterations in PI3KCA were detected in the dataset, discuss the potential implications of this finding. Elaborate on the role of PI3KCA mutations in TNBC and its association with treatment response or clinical outcomes.
Discussion:
a. Compare the findings of the present study with existing literature on the genomic profiles of primary and recurrent TNBC tumors. Highlight any concordant or divergent results, and discuss potential explanations for the observed similarities or differences.
b. Consider the clinical implications of the study findings. How can the identification of consistent mutational profiles between primary and recurrent tumors contribute to TNBC management? Discuss the potential implications for targeted therapy, prognostication, and the development of personalized treatment strategies.
c. Acknowledge the limitations of the study. Address any potential sources of bias or confounding factors that may impact the interpretation of the results. Additionally, discuss any technical limitations or challenges associated with the use of WES in formalin-fixed paraffin-embedded tissues.
Conclusion:
a. Emphasize the key findings of the study, particularly the consistent mutational profiles observed between primary and paired recurrent tumors in TNBC. Highlight the potential clinical relevance and implications for future research in this field.
b. Consider discussing potential future directions for research based on the current findings. Are there any unanswered questions or areas that warrant further investigation? Suggest possible follow-up studies or methodologies that could expand on the current knowledge.
Overall, addressing the above suggestions will strengthen the clarity, scientific rigor, and clinical relevance of the manuscript.
English presentation can be improved
Round 2
Reviewer 1 Report
Accepted
Reviewer 2 Report
I appreciate the effort made by authors in addressing my observations. The manuscript has improved and the authors managed to address my questions. In my view, the paper can be accept.
Review the English presentation again.